# OpenReview forum: "Benchmarking Overton Pluralism in LLMs"
_ICLR.cc/2026/Conference — ICLR 2026 Poster_

### Official Review · Reviewer_6u64 · 2025-10-25

**Soundness:** 2
**Presentation:** 3
**Contribution:** 3
**Rating:** 4
**Confidence:** 3

**Summary:**

The paper introduces OvertonScore, a set-coverage metric for "Overton pluralism", how many distinct, reasonable viewpoints a single model response simultaneously represents on a subjective question. The Overton window W(x) is induced from humans by clustering participants into viewpoint groups via agreement/disagreement judgments. A viewpoint is marked “covered” when its cluster’s mean perceived-representation score for a model response is ≥4/5. Using a U.S.-representative study (N=300; 15 questions) the authors evaluate 8 LLMs, finding adjusted OvertonScores around 0.2–0.37 with o4-mini best on average, well below the 1.0 ceiling. For scalability, they build an automated judge (best: Gemini 2.5 Pro with few-shot + user free-response), which reproduces model rankings with Spearman ρ≈0.88.

**Strengths:**

Human-grounded viewpoint: W(x) comes from participant clustering rather than model outputs

Informative findings: Demonstrates substantial headroom (low scores even at the “best-across-models” upper bound) and a trade-off between neutrality and pluralistic coverage.

Automated evaluation that tracks humans: The judge achieves MAE ≈ 0.66 and ρ ≈ 0.88 on model ranking, making iterative evaluation feasible.

**Weaknesses:**

For benchmark design:

* Benchmark breadth: 15 questions is small for a general benchmark; topical and cultural coverage is limited.

* U.S.-centric panel: Viewpoint discovery and coverage judgments are U.S.-specific; cross-cultural validity is unclear.

For evaluation details:

* The 8 evaluated models may omit newer frontier systems (e.g., GPT-5, Grok-4), which could shift conclusions.

* Differences across chatbot vs. reasoning vs. agentic models are not explored; these may behave differently on pluralism.

* Coverage could increase with higher temperature.

* Overton coverage is inherently prompt-dependent (e.g., prompts that explicitly request multiple perspectives).

**Questions:**

Did you try prompts that explicitly instruct models to “cover multiple reasonable perspectives”? How do scores change?

Can you include newer frontier models to test whether conclusions hold?

Please also see the Weaknesses above.

---

> ### Author Response · Authors · 2025-11-27
>
> We thank the reviewer for their insightful comments. We are delighted they found our **findings informative** and **highlighting the trade-off between neutrality and pluralistic coverage.** We are also glad they recognize how our automated benchmark **matches the human scores** and **enables feasible iterative evaluation.**
>
> We apologize for the late response as we were significantly expanding our data collection and this took some time to complete. Below, we address all comments and have updated the manuscript to incorporate all feedback.
>
>
> 1. Size and breadth
> > Benchmark breadth: 15 questions is small for a general benchmark; topical and cultural coverage is limited.
>
> We appreciate the concern that the current benchmark is based on 15 questions drawn from a US-focused dataset, which may limit generality. Our aim was to establish a methodological foundation using a well-validated dataset (Model Slant) before scaling to additional domains. However, to still address your concern directly, *we have just finished conducting a **substantially expanded round of data collection.***
>
> Specifically, we are increasing the question set **4x**—**from 15 to 60 prompts**—by adding 45 questions from the PRISM dataset “values-guided” subset (Kirk et. al., 2025)—selected using a principled manual screen requiring that questions be subjective, well-formed, elicit genuinely different viewpoints, avoid factual recall or specialized knowledge, and exclude redundant or near-duplicate framings . This expanded set covers subjective topics such as family (e.g., relationship values, parenting approaches), ethics (e.g., honesty, charity), society (e.g., views on hard work, war, globalization), and spirituality (e.g., life’s purpose, existence of aliens, belief in God), among others. We also **increased the participant pool from 300 to 1,209 US-representative respondents** (3 questions each). This increase in sample size ensures greater diversity of viewpoints and more robust estimation of Overton windows across topics, yielding **29,016 datapoints** across 8 widely used LLMs. We updated Section 3 with the expanded dataset information and added a new appendix Section G with more details and the full list of questions (Table 15).
>
> Using this expanded dataset*, we recomputed the full human benchmark (Figure 2, Tables 3–4). While absolute scores increase across models—consistent with PRISM having fewer clusters per question than Model Slant (7.1 vs. 9.6)—the **core conclusion remains unchanged:** models capture only a fraction of the Overton window (theoretical best-across-models upper bound remains similar), and differences across models are modest. Now, no model is statistically significantly above the grand mean on the unweighted OVERTONSCORE, while **Gemma-3-27B** remains significantly below it (p = 0.015). Weighted OVERTONSCOREs remain similarly capped (max 0.53, DeepSeek-V3), comparable to the maximum weighted scores in the original 15-question benchmark (max 0.54, o4-mini). We have updated Section 4.1 with these new results and a new appendix A.1-A.3 discussing model performance on each subset in more detail.
>
> **Cluster quality also remains strong**: the average silhouette score slightly improves (0.380, previously 0.358), and within-cluster agreement / cross-cluster disagreement patterns continue to *indicate well-separated, meaningful viewpoints.* Together, these expanded results show that **our conclusions are not an artifact of the original dataset and that Overton pluralism patterns hold across a much broader and more diverse question set.**
>
> *note that these new results more heavily weight the non-political value laden PRISM topics 3:1. We have added Appendix A.1-A.3 to discuss and compare the results on each dataset.
>
>
> 2. **US focus**
> > U.S.-centric panel: Viewpoint discovery and coverage judgments are U.S.-specific; cross-cultural validity is unclear.
>
> While we agree that measuring pluralism should extend beyond the US, due to resource constraints and to ensure we achieve sufficient statistical power, the scope of this work is US-focused. In theory our methods–both viewpoint discovery (clustering) and coverage judgements (data collection)–**are applicable to all cultures without any modification.** We encourage future work that achieves this and have updated Section 7 to reflect this.

---

> > ### Author Response · Authors · 2025-11-27
> >
> > 3. **Newer models**
> > > The 8 evaluated models may omit newer frontier systems (e.g., GPT-5, Grok-4), which could shift conclusions.
> >
> > > Can you include newer frontier models to test whether conclusions hold?
> >
> > We thank the reviewer for this suggestion. We **incorporated evaluations of three newly released frontier models**—GPT-5.1, Grok-4, and Gemini-3 Pro—using our automated benchmark, which makes rapid preliminary evaluations feasible. The updated results (pasted below as well as in Table 15) show that our **main conclusions remain stable** on the Model Slant questions: o4-mini continues to score highest on Overton pluralism, while Grok-4 and GPT-5.1 also achieve relatively strong coverage and Gemini-3 Pro performs much lower. These additional evaluations both address the reviewer’s concern and further illustrate the **practical value of our automated benchmark for quickly assessing emerging models.** We have created a new Appendix D to include these results.
> >
> > | model                              | adj. coverage |
> > |------------------------------------|---------------|
> > | o4-mini                            | 0.362         |
> > | grok-4                             | 0.348         |
> > | gpt-5.1                            | 0.327         |
> > | deepseek.r1                        | 0.313         |
> > | llama3-3-70b-it              | 0.293         |
> > | gemma-3-27b-it                     | 0.286         |
> > | gpt-4.1                            | 0.272         |
> > | llama-4-maverick | 0.265         |
> > | claude-3-7-sonnet         | 0.230         |
> > | deepseek-v3                        | 0.223         |
> > | gemini-3-pro               | 0.188         |
> >
> > 4. **Chatbot vs reasoning vs agentic models**
> > > Differences across chatbot vs. reasoning vs. agentic models are not explored; these may behave differently on pluralism.
> >
> > We agree that investigating differences across chatbot, reasoning-focused, and agentic models could yield useful insights into how model capabilities interact with pluralistic representation. The current study includes both reasoning (o4-mini, Deepseek R1), hybrid (Claude 3.7 Sonnet), and nonreasoning models (Gemma, Deepseek V3, Llama 3 & 4), enabling a preliminary comparison: our Model Slant benchmark results indicate that **reasoning models outperform nonreasoning models** as evidenced by o4-mini and Deepseek R1 achieving much better OVERTONSCOREs. The hybrid (Claude) and nonreasoning models perform roughly the same. However, on the PRISM questions, we see that Deepseek V3, Llama 3.3 and GPT-4.1 score best–all non-reasoning models. We have **updated the discussion in Section 4.1 to include the above analysis.**
> >
> > With regards to agentic models, unfortunately, a systematic study of this axis is beyond the scope and resource constraints of the current work. We appreciate the suggestion and view this as an important direction for future work, and incorporate it into Section 7.
> >
> > 5. **Temperature and prompt ablations**
> > > Coverage could increase with higher temperature.
> >
> > > Overton coverage is inherently prompt-dependent (e.g., prompts that explicitly request multiple perspectives).
> >
> > > Did you try prompts that explicitly instruct models to “cover multiple reasonable perspectives”? How do scores change?
> >
> > We appreciate the reviewer’s question about temperature and prompting choices. For comparison between benchmarks, we use the actual prompts from other benchmarks which are ecologically valid prompts. In general, we agree that systematically studying how prompting strategies or sampling temperature can enhance pluralistic coverage is valuable future work.
> >
> > For this work, we intentionally use each model’s default prompt and generation settings to measure its baseline behavior when answering value-laden questions—i.e., the behavior users encounter without specialized prompting. This baseline is important because it allows us to compare models on their **intrinsic tendencies toward pluralistic representation**—e.g., how different model types, developers, and training iterations naturally behave before any prompting intervention is applied. As noted above in (4), we find that reasoning models tend more towards pluralistic representation on political domains. Comparing the US- vs. China-based models and the closed vs. open-sourced models, we don’t observe a consistent pattern: o4-mini (US, closed source) and Deepseek R1 (China, open source) both achieve the highest scores, whereas Claude  (US, closed source) and Deepseek V3 (China, open source) both achieve the lowest scores. We have added this to Sec. 4.1 as well.

---

### Official Review · Reviewer_zRFp · 2025-10-29

**Soundness:** 4
**Presentation:** 4
**Contribution:** 4
**Rating:** 10
**Confidence:** 4

**Summary:**

This paper assesses the degree to which LLMs generate responses which faithfully capture the entire range of preferences of human users on political subjects, the Overton window. The define a coverage metric for assessing model pluralism, and score several leading models in a human evaluation. They then extend the human evaluation with an automated assessment that achieves high correlation to human judgements with less overhead.

**Strengths:**

This paper stands out to me as a strong contribution in originality, quality, clarity and significance. The operationalisation of pluralistic alignment via the overton window, and specifically the extent to which surveyed humans agree that their view is represented, is innovative and human-centric. The methods employed are well-described and thorough. The implications for the community are valuable in highlighting the current lack of pluralism across models.

**Weaknesses:**

1. The human sample is limited in size and coverage. Although the authors clearly recognize this, a sample of only 300 people is likely to underrepresent less common views, which are key to an Overton-type analysis. Similarly, the questions and participants are US-centric.
2. There are fundamental trade-offs between Overton coverage and response length. An optimal model probably does not include every possible viewpoint on every topic, which means a perfect score on this benchmark is not always desirable.

**Questions:**

1. Did the participants all complete the study at the same time? How did the early participants get responses to agree/disagree with?
2. Why do you default to the viewpoint-weighted ("unweighted") score rather than the participant-weighted score?
3. How do you propose balancing a desire to hear all views with practical constraints on how long a good response should be?

---

> ### Author Response · Authors · 2025-11-27
>
> We thank the reviewer for their encouraging comments, and are especially delighted they described our work as a **“strong contribution in originality, quality, clarity and significance.”** We are glad they found our operationalization and methodology “innovative and human-centric” and that our **findings have valuable implications for the community.**
>
> We apologize for the late response as we were significantly expanding our data collection and this took some time to complete. Below, we address all comments and have updated the manuscript to incorporate all feedback.
>
> 1. Sample size and breadth
> > The human sample is limited in size and coverage. Although the authors clearly recognize this, a sample of only 300 people is likely to underrepresent less common views, which are key to an Overton-type analysis. Similarly, the questions and participants are US-centric.
>
> We appreciate the concern that the current benchmark is based on 15 questions drawn from a US-focused dataset, which may limit generality. Our aim was to establish a methodological foundation using a well-validated dataset (Model Slant) before scaling to additional domains. However, to still address your concern directly, *we have just finished conducting a **substantially expanded round of data collection.***
>
> Specifically, we are increasing the question set **4x**—**from 15 to 60 prompts**—by adding 45 questions from the PRISM dataset “values-guided” subset (Kirk et. al., 2025)—selected using a principled manual screen requiring that questions be subjective, well-formed, elicit genuinely different viewpoints, avoid factual recall or specialized knowledge, and exclude redundant or near-duplicate framings . This expanded set covers subjective topics such as family (e.g., relationship values, parenting approaches), ethics (e.g., honesty, charity), society (e.g., views on hard work, war, globalization), and spirituality (e.g., life’s purpose, existence of aliens, belief in God), among others. We also **increased the participant pool from 300 to 1,209 US-representative respondents** (3 questions each). This increase in sample size ensures greater diversity of viewpoints and more robust estimation of Overton windows across topics, yielding **29,016 datapoints** across 8 widely used LLMs. We updated Section 3 with the expanded dataset information and added a new appendix Section G with more details and the full list of questions (Table 15).
>
> Using this expanded dataset*, we recomputed the full human benchmark (Figure 2, Tables 3–4). While absolute scores increase across models—consistent with PRISM having fewer clusters per question than Model Slant (7.1 vs. 9.6)—the **core conclusion remains unchanged:** models capture only a fraction of the Overton window (theoretical best-across-models upper bound remains similar), and differences across models are modest. Now, no model is statistically significantly above the grand mean on the unweighted OVERTONSCORE, while **Gemma-3-27B** remains significantly below it (p = 0.015). Weighted OVERTONSCOREs remain similarly capped (max 0.53, DeepSeek-V3), comparable to the maximum weighted scores in the original 15-question benchmark (max 0.54, o4-mini). We have updated Section 4.1 with these new results and a new appendix A.1-A.3 discussing model performance on each subset in more detail.
>
> **Cluster quality also remains strong**: the average silhouette score slightly improves (0.380, previously 0.358), and within-cluster agreement / cross-cluster disagreement patterns continue to *indicate well-separated, meaningful viewpoints.* Together, these expanded results show that **our conclusions are not an artifact of the original dataset and that Overton pluralism patterns hold across a much broader and more diverse question set.**
>
> *note that these new results more heavily weight the non-political value laden PRISM topics 3:1. We have added Appendix A.1-A.3 to discuss and compare the results on each dataset.

---

> > ### Author Response · Authors · 2025-11-27
> >
> > 2. **Length vs coverage**
> > > There are fundamental trade-offs between Overton coverage and response length. An optimal model probably does not include every possible viewpoint on every topic, which means a perfect score on this benchmark is not always desirable.
> >
> > > How do you propose balancing a desire to hear all views with practical constraints on how long a good response should be?
> >
> > We appreciate the reviewer highlighting the important trade-off between response length and viewpoint coverage. Our goal is not to encourage models to enumerate every conceivable perspective regardless of cost or readability; rather, an ideal model will achieve a high score *within practical verbosity constraints*. The best way to compare models is to look for the best score per roughly equal response lengths. Indeed this was the case in our study, where all evaluated model responses are all of comparable length (no model was systematically longer or shorter), and, in our analysis, we did not observe any correlation between response length and coverage nor response length and representation score. This ensures that observed differences in pluralism in our benchmark are not driven by verbosity, but it is important future work to investigate this tradeoff more deeply.
> >
> > More broadly, LLM providers already face strong incentives to keep responses concise for inference cost reasons, so our benchmark can help detect whether efforts to shorten responses inadvertently reduce pluralistic coverage. More generally, our benchmark enables future analyses to find the most practical balance of covering all views without excessive verbosity.
> >
> > > Did the participants all complete the study at the same time? How did the early participants get responses to agree/disagree with?
> >
> > Participants completed the study sequentially so that later participants would be able to vote on the responses of the previous participants. We pre-filled each question voting module with the 10 statements for each question from our pilot study (Appendix G.1) for the early participants to vote on. For the expanded data collection on PRISM, we followed the guidelines in Small et al. (2023) [2] for generating diverse seed statements with LLMs. We appreciate the clarifying question, and have updated these details in Sec. 3 and Appendix C.2.
> >
> > > Why do you default to the viewpoint-weighted ("unweighted") score rather than the participant-weighted score?
> >
> > We present the unweighted score first because it most closely follows the Sorensen et al. 2024 concept of Overton pluralistic alignment, where each distinct viewpoint is treated equally regardless of how many people hold it. The participant-weighted version is motivated by more practical constraints—namely, that a model representing a larger number of people should receive a higher score. This makes it well suited for deployment settings where prevalence matters. In practice, the two metrics capture complementary aspects of model behavior, and our results show that they often diverge in informative ways (see our updated discussion in Section 7 and new analyses in Appendix A). For that reason, we consider both metrics essential and report them side by side rather than treating one as primary.
> >
> > #### References
> >
> > [1] Kirk, H. R., Whitefield, A., Röttger, P., Bean, A., Margatina, K., Ciro, J., Mosquera, R., Bartolo, M., Williams, A., He, H., Vidgen, B., & Hale, S. A. (2024). The PRISM alignment dataset: What participatory, representative and individualised human feedback reveals about the subjective and multicultural alignment of large language models. In Proceedings of the 38th International Conference on Neural Information Processing Systems (NeurIPS 2024) (Vol. 37, Article 3342, pp. 105236–105344). Curran Associates, Inc.
> >
> > [2] Small, C. T., Vendrov, I., Durmus, E., Homaei, H., Barry, E., Cornebise, J., Suzman, T., Ganguli, D., & Megill, C. (2023). Opportunities and risks of LLMs for scalable deliberation with Polis. arXiv:2306.11932

---

### Official Review · Reviewer_Zw8e · 2025-11-02

**Soundness:** 3
**Presentation:** 2
**Contribution:** 2
**Rating:** 4
**Confidence:** 4

**Summary:**

This paper introduces a benchmark for measuring Overton pluralism - the extent to which LLM outputs simultaneously represent multiple reasonable perspectives. The authors formalize OVERTONSCORE as a set-coverage metric, conduct a human study with 300 US participants evaluating 8 LLMs, and develop an automated LLM-as-judge benchmark achieving ρ=0.88 correlation with human judgments.

Novelty. The claim of introducing "the first framework for measuring Overton pluralism" is overstated. Modular Pluralism (Feng et al., 2024) already measures Overton pluralism quantitatively using NLI-based coverage and human win rates, achieving 68.5% improvements. VITAL (Shetty et al., 2025) benchmarks all three pluralism modes including Overton coverage. The genuine novelty lies in the specific methodology: participant voting-based clustering, representation ratings, and human-validated automation. This is a better, more rigorous version of something that exists, not the first of its kind.

Significance. The paper addresses an important problem but overstates its contribution. The claim that methods "are not evaluated directly... due to a lack of benchmarks" contradicts existing work -- Modular Pluralism and VITAL explicitly evaluate Overton pluralism. The actual gap is in standardization and human validation, not evaluation per se. Furthermore, algorithmic monoculture research (Zhang et al., 2025) shows 21 LLMs align with only 41% of human preferences due to fundamental response homogeneity. The paper's findings (0.2-0.37 coverage) confirm this systemic issue, but better benchmarks alone may be insufficient without addressing underlying alignment paradigms. Significance is further limited by: (1) US-only scope, (2) Overton being one of three pluralism types with context-dependent importance, and (3) modest scale compared to PRISM (1,500 participants, 75 countries) and Model Slant (10,007 respondents).

* Shangbin Feng, Taylor Sorensen, Yuhan Liu, Jillian Fisher, Chan Young Park, Yejin Choi, and Yulia Tsvetkov. 2024. Modular Pluralism: Pluralistic Alignment via Multi-LLM Collaboration. In Proceedings of the 2024 Conference on Empirical Methods in Natural Language Processing, pages 4151–4171, Miami, Florida, USA. Association for Computational Linguistics.
* Anudeex Shetty, Amin Beheshti, Mark Dras, and Usman Naseem. 2025. VITAL: A New Dataset for Benchmarking Pluralistic Alignment in Healthcare. In Proceedings of the 63rd Annual Meeting of the Association for Computational Linguistics (Volume 1: Long Papers), pages 22954–22974, Vienna, Austria. Association for Computational Linguistics.
* Taylor Sorensen, Jared Moore, Jillian Fisher, Mitchell Gordon, Niloofar Mireshghallah, Christopher Michael Rytting, Andre Ye, Liwei Jiang, Ximing Lu, Nouha Dziri, Tim Althoff, and Yejin Choi. 2024. Position: a roadmap to pluralistic alignment. In Proceedings of the 41st International Conference on Machine Learning (ICML'24), Vol. 235. JMLR.org, Article 1882, 46280–46302.
* Lily Hong Zhang, Smitha Milli, Karen Jusko, Jonathan Smith, Brandon Amos, Wassim Bouaziz, Manon Revel, Jack Kussman, Yasha Sheynin, Lisa Titus, Bhaktipriya Radharapu, Jane Yu, Vidya Sarma, Kris Rose, Maximilian Nickel (2025). Cultivating Pluralism In Algorithmic Monoculture: The Community Alignment Dataset. arXiv:2507.09650

**Strengths:**

-  Novel clustering methodology: Using participant voting patterns to identify viewpoints is innovative and more faithful to human understanding than semantic similarity or NLI approaches.
- Rigorous statistical framework: Proper significance testing with OLS models, fixed effects, and cluster-robust standard errors enables principled model comparison.
- Validated automation: LLM-as-judge achieves strong correlation (ρ=0.88) with human judgments and shows small fairness disparities (η² < 0.004), providing practical utility for iterative development.
- Confirms systemic gap: Models achieve only 0.2-0.37 coverage (even pooled: 0.62), consistent with broader algorithmic monoculture findings showing fundamental limitations in current LLMs.
- Clear formalization: Set-coverage metric provides intuitive operationalization with both weighted and unweighted variants.
- Actionable insights: Identifies o4-mini as significantly above average (p=0.043) and DeepSeek V3 below (p=0.017).

**Weaknesses:**

- False "first" claim: Modular Pluralism and VITAL already measure Overton pluralism. The contribution is methodological refinement (human validation + clustering), not pioneering measurement.
- Contradictory "lack of benchmarks" statement: Page 1 claims methods aren't evaluated due to lacking benchmarks, but Modular Pluralism explicitly evaluates Overton pluralism improvements.
- Limited scope: US-only, 15 questions, 300 participants. PRISM has 75 countries; Model Slant has 10,007 respondents; Community Alignment has 15,000 across 5 countries. Overton windows are culturally situated -- generalizability is severely limited.
- Modest clustering quality: Silhouette score of 0.358 indicates only moderate separation, questioning whether "distinct viewpoints" are genuine or algorithmic artifacts.
- Unclear practical importance: Overton pluralism is one of three types (Sorensen et al., 2024b note distributional/steerable get more attention). Context-dependent value not established.
- May address wrong problem: Algorithmic monoculture research shows root cause is response homogeneity in alignment processes, not measurement inadequacy. Better benchmarks alone may be insufficient.
- Question set overlap with Model Slant: o4-mini ranks most pluralistic here but second-most politically slanted there. This relationship needs investigation -- potential confounds from verbosity/style not addressed.
- Automated benchmark limitations: Systematic over-rating of Claude 3.7 Sonnet (+0.104 in Table 1) and significant subgroup differences suggest generalization issues.
- Binary coverage threshold: Rating ≥4 = covered is arbitrary; no sensitivity analysis provided.
- Missing ablations: No analysis of how coverage varies by question difficulty, cluster size, or agreement levels.

**Questions:**

- How does your representation-rating approach differ substantively from Modular Pluralism's human win-rate evaluation for Overton pluralism?
- Can you provide systematic comparison between your OVERTONSCORE rankings and Model Slant rankings? Do high pluralism and perceived slant correlate? Could verbosity/hedging confound both?
- How sensitive are rankings to clustering hyperparameters (silhouette=0.358)? Did you validate clusters qualitatively?
- How do results change with coverage thresholds of 3.5 or 4.5 instead of 4.0?
- Given algorithmic monoculture research showing fundamental response homogeneity limits preference learning even with diverse datasets, how does better measurement address this? Would your benchmark be more useful for evaluating data collection strategies than model selection?
- Best-across-models achieves only 0.62 -- does this suggest ensemble approaches or that single pluralistic models are infeasible?
- How do you distinguish valuable pluralism from harmful false balance when some viewpoints are epistemically inferior?
- What is computational cost vs. human evaluation? How frequently should automated benchmarks be re-validated?
- Do you expect same viewpoint clusters cross-culturally? How would you adapt methodology for non-US populations?
- Which metric should be primary -- weighted or unweighted? How prevent gaming where models optimize for largest clusters while ignoring minorities?

---

> ### Author Response · Authors · 2025-11-27
>
> We thank Reviewer Zw8e so much for the detailed and deeply insightful feedback! We are delighted they describe our methodology as **“innovative and more faithful to human understanding than [other] approaches”** and recognize our contributions as “a **better, more rigorous version** of something that exists.” We are also encouraged they found our operationalization of Overton pluralism *clear and intuitive*, appreciated the **statistical rigor** of our work, and how our results provide **actionable insights and practical utility.**
>
> We apologize for the late response as we were significantly expanding our data collection and this took some time to complete. Below, we address all comments and have updated the manuscript to incorporate all feedback.
>
> 1. **Novelty & Significance**
> > False "first" claim: Modular Pluralism and VITAL already measure Overton pluralism. The contribution is methodological refinement (human validation + clustering), not pioneering measurement.
>
> > Contradictory "lack of benchmarks" statement: Page 1 claims methods aren't evaluated due to lacking benchmarks, but Modular Pluralism explicitly evaluates Overton pluralism improvements.
>
> We thank the reviewer for the very helpful reframing. We fully agree that the genuine novelty of our work lies not in being first but in *how* we operationalize Overton pluralism: (i) discovering viewpoints through participant voting patterns rather than NLI or prompt-based clustering; (ii) grounding coverage in direct human representation ratings; and (iii) validating an automated judge against the human benchmark. **We appreciate the reviewer highlighting this as “a better, more rigorous version of something that exists,”** and we have revised the Introduction to reflect this more accurate positioning.
>
> Prior work such as Modular Pluralism and VITAL each include an Overton component, but they approach it very differently than in our work. Modular Pluralism and VITAL both do (i) **NLI-based value detection** using the Value Kaleidoscope dataset, and (ii) **pairwise response win-rate** evaluations where human/GPT-4 annotators choose which response is more pluralistic. These methods do not estimate the Overton window itself nor measure coverage over distinct human viewpoints; instead, they test whether one model output appears better than another or whether it entails predefined values. By contrast, our benchmark (i) **discovers viewpoints directly from humans** through agreement/disagreement voting, (ii) **tests coverage using perceived representation ratings from the people who hold each viewpoint**, and (iii) calculates a **set-coverage metric** aligned with the formal definition of Overton pluralism. In other words, our method does not assume a fixed value taxonomy nor rely on entailment heuristics; it measures whether real participants feel represented by a model’s answer. We have revised the Introduction to clarify this distinction and avoid overclaiming novelty, incorporating the reviewer’s more accurate framing that our contribution is a more rigorous, human-grounded operationalization and benchmark.
>
> > How does your representation-rating approach differ substantively from Modular Pluralism's human win-rate evaluation for Overton pluralism?
>
> To explicitly address your question, Modular Pluralism’s human win-rate (pairwise choice of which model is more pluralistic) does not identify *which* viewpoints exist nor measure *coverage* of them; it simply ranks two responses. In contrast, our representation-rating approach identifies human viewpoint clusters for each question and directly measures whether members of each viewpoint feel represented by a model’s output. This yields a set-coverage metric over the empirical Overton window itself (+ enabling analyses into which viewpoints specifically are covered or missed), not a relative preference between model responses.

---

> > ### Author Response · Authors · 2025-11-27
> >
> > 2. **Scope**
> > > Limited scope: US-only, 15 questions, 300 participants. PRISM has 75 countries; Model Slant has 10,007 respondents; Community Alignment has 15,000 across 5 countries. Overton windows are culturally situated -- generalizability is severely limited.
> >
> > We appreciate the concern that the current benchmark is based on 15 questions drawn from a US-focused dataset, which may limit generality. Our aim was to establish a methodological foundation using a well-validated dataset (Model Slant) before scaling to additional domains. However, to still address your concern directly, *we have just finished conducting a **substantially expanded round of data collection.***
> >
> > Specifically, we are increasing the question set **4x**—**from 15 to 60 prompts**—by adding 45 questions from the PRISM dataset “values-guided” subset (Kirk et. al., 2025)—selected using a principled manual screen requiring that questions be subjective, well-formed, elicit genuinely different viewpoints, avoid factual recall or specialized knowledge, and exclude redundant or near-duplicate framings . This expanded set covers subjective topics such as family (e.g., relationship values, parenting approaches), ethics (e.g., honesty, charity), society (e.g., views on hard work, war, globalization), and spirituality (e.g., life’s purpose, existence of aliens, belief in God), among others. We also **increased the participant pool from 300 to 1,209 US-representative respondents** (3 questions each). This increase in sample size ensures greater diversity of viewpoints and more robust estimation of Overton windows across topics, yielding **29,016 datapoints** across 8 widely used LLMs. We updated Section 3 with the expanded dataset information and added a new appendix Section G with more details and the full list of questions (Table 15).
> >
> > Using this expanded dataset*, we recomputed the full human benchmark (Figure 2, Tables 3–4). While absolute scores increase across models—consistent with PRISM having fewer clusters per question than Model Slant (7.1 vs. 9.6)—the **core conclusion remains unchanged:** models capture only a fraction of the Overton window (theoretical best-across-models upper bound remains similar), and differences across models are modest. Now, no model is statistically significantly above the grand mean on the unweighted OVERTONSCORE, while **Gemma-3-27B** remains significantly below it (p = 0.015). Weighted OVERTONSCOREs remain similarly capped (max 0.53, DeepSeek-V3), comparable to the maximum weighted scores in the original 15-question benchmark (max 0.54, o4-mini). We have updated Section 4.1 with these new results and a new appendix A.1-A.3 discussing model performance on each subset in more detail.
> >
> > **Cluster quality also remains strong**: the average silhouette score slightly improves (0.380, previously 0.358), and within-cluster agreement / cross-cluster disagreement patterns continue to *indicate well-separated, meaningful viewpoints.* Together, these expanded results show that **our conclusions are not an artifact of the original dataset and that Overton pluralism patterns hold across a much broader and more diverse question set.**
> >
> > *note that these new results more heavily weight the non-political value laden PRISM topics 3:1. We have added Appendix A.1-A.3 to discuss and compare the results on each dataset.
> >
> > 3. **Clustering quality**
> > > Modest clustering quality: Silhouette score of 0.358 indicates only moderate separation, questioning whether "distinct viewpoints" are genuine or algorithmic artifacts.
> >
> > We appreciate the concern regarding the clustering separation, and agree that silhouette scores should be interpreted cautiously. In our case, we are working with high-dimensional sparse-vote matrices (as with Small et al., 2021 [2]). As noted in classical results on the curse of dimensionality, it becomes very difficult to achieve high silhouette values (Beyer et al., 1999 [3]). In this context, our mean silhouette reflects meaningful separation between viewpoint clusters.
> >
> > Moreover, to further quantify the extent to which clusters represent meaningful viewpoints, we now additionally calculate a cluster cohesion score for each cluster based on the peer-voting matrix. We define within-cluster as the fraction of within-cluster votes in which participants agreed with statements written by other participants in the same cluster. We find that the average cohesion across all clusters (excluding singletons where it’s trivially 1.0) is very high at $0.86$, indicating that clusters reflect highly coherent viewpoints as intended.

---

> > > ### Author Response · Authors · 2025-11-27
> > >
> > > To contextualize this number, we additionally compare within-cluster and out-of-cluster voting behavior. For each cluster $C$, we compute the proportion of times members of $C$ approve (cohesion), disapprove, or pass on statements authored by other members of $C$, and compare this to their voting on statements authored by individuals outside $C$. Averaged across all questions, **within-cluster approval is substantially higher ($0.849$) than out-of-cluster approval ($0.490$),** while within-cluster disapproval remains extremely low ($0.058$) compared to out-of-cluster disapproval ($0.377$). These results show that **our clusters reflect genuine differences in perspective rather than noise or algorithmic artifacts,** providing strong evidence of the validity of our clustering procedure as a means of identifying distinct viewpoints. We have added this line to Sec. 4 and include full details in a new Appendix C.3 & Table 13.
> > >
> > > 4. **Practical Importance**
> > > > Unclear practical importance: Overton pluralism is one of three types (Sorensen et al., 2024b note distributional/steerable get more attention). Context-dependent value not established.
> > >
> > > We appreciate the chance to clarify scope. Our benchmark is **intentionally focused on Overton pluralism**, one of the three pluralism modes identified by Sorensen et al. (2024b). A single benchmark cannot measure all modes simultaneously, and our goal is to deliver a high-fidelity, human-validated operationalization of this specific construct. The reviewer’s own strengths notes that our automated benchmark “provides practical utility for iterative development,” which we have expanded upon in Appendix F to show concrete ways developers can use both the human and automated benchmark in model-development cycles.
> > >
> > > 5. **Algorithmic monoculture**
> > > > May address wrong problem: Algorithmic monoculture research shows root cause is response homogeneity in alignment processes, not measurement inadequacy. Better benchmarks alone may be insufficient.
> > >
> > > > Given algorithmic monoculture research showing fundamental response homogeneity limits preference learning even with diverse datasets, how does better measurement address this? Would your benchmark be more useful for evaluating data collection strategies than model selection?
> > >
> > > We agree that response homogeneity is a fundamental alignment challenge. Our results **empirically corroborate monoculture effects** (coverage of 0.2–0.4; pooled maximum 0.69). Benchmarks alone cannot solve monoculture, but they do *enable progress*by making it *measurable. Our work complements monoculture-focused research by offering a rigorous way to **diagnose (the current lack of) pluralistic coverage** and **track improvements** across both alignment and/or dataset collection strategies.
> > >
> > > 6. **Model Slant**
> > > > Question set overlap with Model Slant: o4-mini ranks most pluralistic here but second-most politically slanted there. This relationship needs investigation -- potential confounds from verbosity/style not addressed.
> > >
> > > We thank the reviewer for raising this point. Interestingly, our results uncover a substantive finding: o4-mini ranks as the second most politically slanted model in Model Slant, yet is by far the most Overton-pluralistic in our benchmark. This underscores our motivation for including the Model Slant questions in our study: our approach enables more deeply understanding whether any model slant could be due to perspective exclusion (e.g. a politically neutral response excludes many perspectives) versus biased inclusion (e.g. a model includes many different perspectives, but is weighted more towards one view).
> > > **We discuss this in the revised manuscript** in Section 8, as it suggests a potential trade-off between neutrality-as-low-slant and pluralistic representation. This divergence further motivates the need for a dedicated Overton pluralism metric.

---

> > > > ### Author Response · Authors · 2025-11-27
> > > >
> > > > > Can you provide systematic comparison between your OVERTONSCORE rankings and Model Slant rankings? Do high pluralism and perceived slant correlate?
> > > >
> > > > We appreciate the insightful question. Using the Model Slant dataset’s primary metric (overall perceived slant, where values closer to 0 indicate greater neutrality), **we correlated Model Slant rankings with our OVERTONSCOREs.** Across the seven models appearing in both datasets (Model Slant doesn’t include Deepseek V3), we find a **moderate negative association** between the two measures (Pearson $r = −0.41$; Spearman $\rho = −0.32$; Kendall $\tau = −0.24$). In other words, models judged as more politically neutral under Model Slant are less pluralistic in our benchmark. This divergence supports our motivation for including Model Slant questions in our study and demonstrates that political slant and Overton pluralism measure *distinct* phenomena. Neutral-sounding responses may achieve low perceived slant by collapsing or omitting minority viewpoints, whereas pluralistic responses may surface multiple perspectives and thus be perceived as more “biased” in pairwise slant comparisons. We include a **new Appendix B with a detailed analysis on this comparison.**
> > > >
> > > > > Could verbosity/hedging confound both?
> > > >
> > > > In our work, we found no correlation between the verbosity (response length) and representation ratings, partly due to the very similar response lengths across models. However large discrepancies in response lengths would certainly affect the degree of pluralism in other cases. We agree that ablations testing directly for length and analysis of hedging or other linguistic factors is valuable future work, and we have added this to the discussion (Sec. 7).
> > > >
> > > > 7. **Generalization**
> > > > > Automated benchmark limitations: Systematic over-rating of Claude 3.7 Sonnet (+0.104 in Table 1) and significant subgroup differences suggest generalization issues.
> > > >
> > > > We acknowledge the systematic over-rating of Claude 3.7 Sonnet. Importantly, the judge still **preserves relative rankings at practical decision thresholds** (Precision@2 = 1/2; @4 = 3/4; @6 = 5/6). In realistic development loops—where automated evaluation narrows candidates before a final round of human assessment—such deviations would **not harm** selection fidelity. Moreover, concurrent work (e.g., Kolluri et al., 2025 [4]) shows that fine-tuning LLM judges for behavioral prediction is feasible, suggesting clear avenues for improving judge robustness.
> > > >
> > > > 8. **Threshold sensitivity analysis**
> > > > > Binary coverage threshold: Rating ≥4 = covered is arbitrary; no sensitivity analysis provided.
> > > >
> > > > We agree that such analyses are important, and therefore **we performed a sensitivity analysis** across five values ($\tau \in [3.6, 3.7, 3.8, 3.9, 4.0]$). **Model rankings were highly stable** (median Kendall $\tau = 0.64$ unweighted, 0.71 weighted), and the **top-3 models remained unchanged across all thresholds.** For the Model Slant subset, these values are even stronger:  median Kendall $\tau = 0.84$ unweighted, 0.93 weighted and top model is o4-mini 100% of the time. For the PRISM subset, we similarly see median Kendall $\tau = 0.93$ unweighted, 0.86 weighted and top-2 models are consistent across thresholds. We will include full results and pairwise win-rate consistency heatmaps in Appendix A.6.
> > > >
> > > > 9. **Ablations**
> > > > > Missing ablations: No analysis of how coverage varies by question difficulty, cluster size, or agreement levels.
> > > >
> > > > We thank the reviewer for these helpful ablations. We have now **conducted three complementary analyses** addressing (A) question difficulty, (B) cluster-size effects, and (C) within-cluster agreement levels. All results are included in the revised Appendix, and we present an overview below:
> > > >
> > > >     A. **Coverage vs. Question Difficulty (number of clusters)**
> > > > We thank the reviewer for this helpful suggestion. We define question difficulty as the number of distinct viewpoints $K_x$, since questions with many clusters present a wider Overton window that could, in principle, be harder for models to cover.  Across all models and questions, the correlation between $K_x$ and per-question coverage is $r=-0.17$. These results indicate that **models maintain broadly consistent coverage** even on questions with many distinct viewpoints and reinforces the robustness of the benchmark. We have added the detailed ablation results to Appendix A.4 (Table 9).

---

> > > > > ### Author Response · Authors · 2025-11-27
> > > > >
> > > > > B. **Coverage vs. Cluster Size**
> > > > >
> > > > > As per your suggestion, we compute the correlation between cluster size and its mean representation rating across all models and questions. We find a pooled correlation of $r=0.249$, indicating a **weak tendency** for larger clusters to have higher average representation ratings. This confirms that the unweighted **metric is not dominated by cluster size** (i.e. not biased toward majority viewpoints) and that reporting both weighted and unweighted metrics provides a more complete picture of model behavior. We have added these detailed results to Appendix A.5 (Table 10).
> > > > >
> > > > >
> > > > >     C. **Coverage vs. Within-Cluster Cohesion (Agreement Levels)**
> > > > > To address the question about agreement levels, **correlate the within-cluster cohesion score** as described in (3) **with each cluster’s mean representation rating** across models. We find the overall pooled correlation is $r=0.006$, i.e. **no association**. This suggests that representation ratings (and therefore coverage) are **independent of viewpoint cohesion**, and that the **coverage scores meaningfully capture the extent to which models represent each viewpoint**, not just reflecting agreement levels. We have added the detailed ablation results to Appendix C.3.
> > > > >
> > > > > ### Questions
> > > > > Below, we address the questions not covered above.
> > > > >
> > > > > > Best-across-models achieves only 0.62 -- does this suggest ensemble approaches or that single pluralistic models are infeasible?
> > > > >
> > > > > No. This upper bound reflects the *theoretical* best combination of each model by taking the best cluster scores across models. In the future, we are optimistic that either single-model or ensemble approaches can achieve much higher pluralistic representation.
> > > > >
> > > > > > How do you distinguish valuable pluralism from harmful false balance when some viewpoints are epistemically inferior?
> > > > >
> > > > > In the scope of this work, we chose *subjective* questions deliberately such that all answers are epistemically valid. We agree that such a distinction is important in other domains and it is valuable future work.
> > > > >
> > > > > > What is computational cost vs. human evaluation? How frequently should automated benchmarks be re-validated?
> > > > >
> > > > > The human evaluation cost for the 60 questions and 1200 participants via Prolific was US \$8400 (average hourly rate of US \$12-13). The computational requirements for the predictions with Gemini 2.5 Pro via API are very cheap, done entirely on a CPU. In terms of API costs, we estimate roughly US\$30 for the final prediction run (though likely less due to prompt caching on the Google API side).
> > > > >
> > > > > In terms of re-validation, we recommend to re-validate the predictions with humans at least at the final stage of model development when selecting the best version for deployment. Moreover, one could run a more lightweight version of the human evaluation (e.g. just to validate the predictions on a selected subset of models, instead of collecting all ratings from scratch on all models) at significant decision points during the model development pipeline. We have added Appendix F giving a practical example of how our benchmark fits into the model development cycle.
> > > > >
> > > > > > Do you expect same viewpoint clusters cross-culturally? How would you adapt methodology for non-US populations?
> > > > >
> > > > > In the case of questions where other cultures would hold different viewpoints than the US, we would expect to see both the current US-based viewpoint clusters and any new ones from the other cultures. **Our methodology would not need to be adapted and is applicable to any global population(s).** Within the academic resources available to us and to ensure we achieve sufficient statistical power, the scope of this work is US-focused. We agree that extending to more diverse global populations is critical future research and we have updated Section 7 to reflect this.
> > > > >
> > > > > > Which metric should be primary -- weighted or unweighted? How prevent gaming where models optimize for largest clusters while ignoring minorities?
> > > > >
> > > > > As suggested in the paper and in (9B) above, **the weighted and unweighted metrics capture complementary aspects of Overton pluralism**, and thus discuss both in our benchmark. However, there may be certain situations in which one may be more relevant, but this would depend on the specific context and use-case. In regards to over optimizing for the largest clusters, our unweighted metric is exactly fit for this scenario because all clusters contribute equally to the score regardless of size. Moreover, one could also detect the presence of this over optimization exactly by comparing the weighted and unweighted metrics: a substantially larger weighted vs unweighted score would be evidence of this.

---

> > > > > > ### Author Response · Authors · 2025-11-27
> > > > > > **References**
> > > > > >
> > > > > > [1] Kirk, H. R., Whitefield, A., Röttger, P., Bean, A., Margatina, K., Ciro, J., Mosquera, R., Bartolo, M., Williams, A., He, H., Vidgen, B., & Hale, S. A. (2024). The PRISM alignment dataset: What participatory, representative and individualised human feedback reveals about the subjective and multicultural alignment of large language models. In Proceedings of the 38th International Conference on Neural Information Processing Systems (NeurIPS 2024) (Vol. 37, Article 3342, pp. 105236–105344). Curran Associates, Inc.
> > > > > >
> > > > > > [2] Small, C., Bjorkegren, M., Erkkilä, T., Shaw, L., & Megill, C. (2021). Polis: Scaling Deliberation by Mapping High Dimensional Opinion Spaces. RECERCA. Revista de Pensament i Anàlisi, 26(2). doi: 10.6035/recerca.5516
> > > > > >
> > > > > > [3] Beyer, K., Goldstein, J., Ramakrishnan, R., Shaft, U. (1999). When Is “Nearest Neighbor” Meaningful?. In: Beeri, C., Buneman, P. (eds) Database Theory — ICDT’99. ICDT 1999. Lecture Notes in Computer Science, vol 1540. Springer, Berlin, Heidelberg. doi: 10.1007/3-540-49257-7_15.
> > > > > >
> > > > > > [4] Kolluri, A., Wu, S., Park J. S., and Bernstein, M. S. (2025). Finetuning LLMs for Human Behavior Prediction in Social Science Experiments. In Proceedings of the 2025 Conference on Empirical Methods in Natural Language Processing, pages 30084–30099, Suzhou, China. Association for Computational Linguistics.

---

### Official Review · Reviewer_fPY8 · 2025-11-04

**Soundness:** 2
**Presentation:** 3
**Contribution:** 1
**Rating:** 2
**Confidence:** 3

**Summary:**

The authors proposed methods to benchmark Overton pluralism, measuring the proportion of represented perspectives in LLM responses. They conducted a user study, which resulted in the development of an automated evaluation of overtoon pluralism that can be used for model development, as it highly correlates with human data.

**Strengths:**

- The paper is well-presented and explained, outlining the concept of Overton pluralism and how it is measured and evaluated.

- The paper is accompanied by a dataset, where users were asked to evaluate statements in free form or from selected views, as well as rate models’ responses. I believe that the experimental protocol is thoughtfully designed for the narrow setting it targets: participants provide both free-form statements and Likert ratings, as well as pairwise agreement votes, which are then used for clustering.

**Weaknesses:**

- The study’s focus is very narrow and is actually dataset and task-dependent. It does not tell us much about the model’s abilities or how these scores can be generalised or used for developing models. The entire benchmark is built on 15 questions, drawn from a US-focused political dataset.

- I believe that the paper’s contribution is limited for this type of conference, and can be better suited for a workshop.

- The automated benchmark is presented as a tool for model selection, but the paper only demonstrates correlation with human results. It does not demonstrate a concrete development loop where this metric actually guides model improvement or selection.

**Questions:**

I do not have questions for the authors, as they have clearly stated the limitations of their work.

---

> ### Author Response · Authors · 2025-11-27
>
> We thank you for your time and insightful feedback. We are especially glad to hear that you found our **experimental design to be thoughtful** and that our operationalization of Overton pluralism is **well-presented.**
>
> We apologize for the late response as we were significantly expanding our data collection and this took some time to complete. We respond to your comments below and incorporate all feedback in the revised manuscript.
>
> 1. **Study / Dataset Breadth**
> > The study’s focus is very narrow and is actually dataset and task-dependent. It does not tell us much about the model’s abilities or how these scores can be generalised or used for developing models. The entire benchmark is built on 15 questions, drawn from a US-focused political dataset.
>
> We appreciate the concern that the current benchmark is based on 15 questions drawn from a US-focused dataset, which may limit generality. Our aim was to establish a methodological foundation using a well-validated dataset (Model Slant) before scaling to additional domains. However, to still address your concern directly, *we have just finished conducting a **substantially expanded round of data collection.***
>
> Specifically, we are increasing the question set **4x**—**from 15 to 60 prompts**—by adding 45 questions from the PRISM dataset “values-guided” subset (Kirk et. al., 2025)—selected using a principled manual screen requiring that questions be subjective, well-formed, elicit genuinely different viewpoints, avoid factual recall or specialized knowledge, and exclude redundant or near-duplicate framings . This expanded set covers subjective topics such as family (e.g., relationship values, parenting approaches), ethics (e.g., honesty, charity), society (e.g., views on hard work, war, globalization), and spirituality (e.g., life’s purpose, existence of aliens, belief in God), among others. We also **increased the participant pool from 300 to 1,209 US-representative respondents** (3 questions each). This increase in sample size ensures greater diversity of viewpoints and more robust estimation of Overton windows across topics, yielding **29,016 datapoints** across 8 widely used LLMs. We updated Section 3 with the expanded dataset information and added a new appendix Section G with more details and the full list of questions (Table 15).
>
> Using this expanded dataset*, we recomputed the full human benchmark (Figure 2, Tables 3–4). While absolute scores increase across models—consistent with PRISM having fewer clusters per question than Model Slant (7.1 vs. 9.6)—the **core conclusion remains unchanged:** models capture only a fraction of the Overton window (theoretical best-across-models upper bound remains similar), and differences across models are modest. Now, no model is statistically significantly above the grand mean on the unweighted OVERTONSCORE, while **Gemma-3-27B** remains significantly below it (p = 0.015). Weighted OVERTONSCOREs remain similarly capped (max 0.53, DeepSeek-V3), comparable to the maximum weighted scores in the original 15-question benchmark (max 0.54, o4-mini). We have updated Section 4.1 with these new results and a new appendix A.1-A.3 discussing model performance on each subset in more detail.
>
> **Cluster quality also remains strong**: the average silhouette score slightly improves (0.380, previously 0.358), and within-cluster agreement / cross-cluster disagreement patterns continue to *indicate well-separated, meaningful viewpoints.* Together, these expanded results show that **our conclusions are not an artifact of the original dataset and that Overton pluralism patterns hold across a much broader and more diverse question set.**
>
> *note that these new results more heavily weight the non-political value laden PRISM topics 3:1. We have added Appendix A.1-A.3 to discuss and compare the results on each dataset.

---

> > ### Author Response · Authors · 2025-11-27
> >
> > 2. **Use for model development**
> >
> > > The automated benchmark is presented as a tool for model selection, but the paper only demonstrates correlation with human results. It does not demonstrate a concrete development loop where this metric actually guides model improvement or selection.
> >
> > We agree that an explicit demonstration of a model-development loop would further clarify how the automated benchmark supports model selection, and have **added a more concrete description of this iterative loop in Appendix F.** Our intended workflow mirrors existing automated alignment pipelines: a developer training several candidate models (e.g., with different SFT mixtures or post-training objectives) can run our automated benchmark after each iteration to rapidly identify the most promising variants, and only advance those candidates to a full human evaluation. The automated benchmark preserves the relative rankings of the top-performing models under the automated metric compared to the human ratings. In our results, this **selection fidelity is strong at practical decision thresholds**: the automated benchmark preserves 50% of the human top-2 models, 75% of the top-4 models, and 83% of the top-6 models (Precision@2 = 1/2; Precision@4 = 3/4; Precision@6 = 5/6). This shows that a developer could run many inexpensive fine-tuning or RLHF variants, keep only the top-ranked runs according to the automated OvertonScore, and reserve human evaluation for those few—precisely the workflow used in typical development settings.

---

### Author Response · Authors · 2025-12-02

Dear Area Chair,

We were disappointed to hear about the modified policy, as we were optimistic that our replies had significantly addressed the reviewers' concerns and that further engagement likely would have elicited higher scores.

In case it is helpful, we would like to offer a summary of the replies, the changes we have made, and the state of the discussion.

* ﻿***All four reviewers found our innovative methodology to be a major strength:***
  * Zw8e: “The **genuine novelty lies in the specific methodology**” which is ”**a better, more rigorous version of something that exists**” and our “**novel clustering methodology**... **is innovative and more faithful to human understanding**" than in prior work
  * zRFp: “This paper stands out to me as a **strong contribution in originality, quality, clarity and significance**” and our “**operationalisation… is innovative and human-centric**" and “**methods** employed are well-described and **thorough**”
  * fPY8: "experimental protocol is **thoughtfully designed**."
  * 6u64: ‘**Human-grounded viewpoint**” is one of our paper’s strengths because “W(x) comes from participant clustering rather than model outputs”
* ﻿***Three reviewers found our automated benchmark and findings informative and practically useful:***
  * Zw8e: Our “validated automation… providing **practical utility** for iterative development” and our findings “**confirms systemic gap**” and provides “**actionable insights**.”
  * zRFp: “The **implications for the community are valuable** in **highlighting the current lack of pluralism across models**”
  * 6u64: “**Informative findings**: **Demonstrates substantial headroom…** and a **trade-off between neutrality and pluralistic coverage**” and **making iterative evaluation feasible**”
* ﻿***We made significant changes to improve the paper:***
  * We collected a *substantial* amount of new data, **increasing the question set 4x—from 15 to 60 prompts**—and **increased the participant pool from 300 to 1,209 US-representative respondents**, yielding 29,016 datapoints across 8 widely used LLMs. Using this expanded dataset, we recomputed the full human benchmark and our **core conclusion remains unchanged**: models capture only a fraction of the Overton window.
    * All four reviewers noted that our initial dataset was too small, and this expansion **strengthens our results** and ensures **greater diversity of viewpoints** and **more robust estimation of Overton windows** across topics.
  * ﻿Additional robustness checks for our benchmark requested by Reviewer Zw8e.
    * Clustering quality: Averaged across all questions, **within-cluster approval is substantially higher ($0.849$)** than out-of-cluster approval ($0.490$). These results show that **our clusters reflect genuine differences in perspective** rather than noise or algorithmic artifacts, providing **strong evidence of the validity of our clustering procedure** and complementing the Silhouette scores. (Appendix C.3 & Table 13)
    * We performed a **threshold sensitivity analysis** across five values. Model rankings were **highly stable** (median Kendall $tau = 0.64$ unweighted, 0.71 weighted), and the top-3 models remained unchanged across all thresholds.
    * Three complementary analyses addressing (A) question difficulty (Appendix A.4, Table 9), (B) cluster-size effects (Appendix A.5, Table 10), and (C) within-cluster agreement levels (Appendix C.3).  **All these experiments significantly strengthen our results.**
  * As per the suggestion by Reviewer 6u64, we additionally **evaluated three newly released frontier models—GPT-5.1, Grok-4, and Gemini-3 Pro**—using our automated benchmark. The updated results (Appendix D, Table 15) show that our main conclusions remain stable. These additional evaluations **both address the reviewer’s concern and further illustrate the practical value of our automated benchmark for quickly assessing emerging models**.
  * To address Zw8e’s question on novelty: **we provide an in-depth comparison of our work to Modular Pluralism and VITAL**’s estimation of Overton pluralism. These methods do not estimate the Overton window itself nor measure coverage over distinct human viewpoints. By contrast, our benchmark (i) discovers viewpoints directly from humans, (ii) tests coverage using perceived representation ratings, and (iii) calculates a novel metric aligned with the formal definition of Overton pluralism.
  * ﻿While many reviewers praised the clarity of our paper, we made additional improvements to the paper content and presentation, including a Figure 1 overview, improved flow, added additional details to related works and discussion, several new appendices, and more.

We believe that the reviewers all found the problem we are studying to be important and of interest, and believe that we have significantly addressed the stated concerns. Thank you for your consideration of our paper amidst the circumstances.

---

### Meta-Review · Area_Chair_cYDM · 2026-01-08

**Summary:**

I’d lean accept because the submission reads as a solid measurement/benchmark paper with (i) a clear formalization (OVERTONSCORE), (ii) a large-scale U.S.-representative human study (N=1209; 60 questions; 8 LLMs) and (iii) an automated proxy that tracks human rankings well (Spearman ρ≈0.88)—together making pluralistic alignment measurable and reproducible.

The critical concerns across reviews were: scope/generalizability (initially too narrow / U.S.-centric), novelty positioning vs prior pluralism work, and whether the automated benchmark is more than “correlation” (i.e., whether it supports practical iteration). Secondary concerns included cluster validity (silhouette/interpretability), arbitrary coverage threshold, and judge artifacts (systematic over/under-rating of specific models).

**Reviewer Concerns:**

**Largely addressed**

“Too narrow” dataset / too few prompts: expanded to 60 prompts (15 Model Slant + 45 PRISM values-guided) and N=1209 U.S.-based participants, materially reducing the “15 questions” critique.

Evidence that the benchmark yields nontrivial findings across domains: results are broken out by Model Slant vs PRISM; notably, DeepSeek V3 is significantly above-average on PRISM while o4-mini is significantly above-average on Model Slant, and DeepSeek V3 is significantly below-average on Model Slant—showing the benchmark is sensitive rather than trivial.


Automated evaluation practicality beyond “just correlation”: they provide evidence the judge approximates human OVERTONSCOREs closely on a subset and discuss using it for early selection/iteration; they also explicitly note one major failure mode (Claude 3.7 over-rated).


Threshold arbitrariness / robustness: the paper points to a threshold sensitivity analysis and includes additional clustering-quality analyses in appendices.


Model coverage freshness: they extend the automated benchmark to newly released frontier models (GPT-5.1, Grok-4, Gemini 3 Pro), which directly addresses “models are old / missing newer systems.”


**Still outstanding**

Generalizability is still U.S.-bounded (and English-speaking participants): even with broader topics, the Overton window is culturally situated; the current benchmark is still fundamentally U.S.-sampled.

Novelty framing is improved but not fully clean: despite acknowledging related directions, the manuscript still uses “first framework” style language in places, which may continue to irritate a well-read pluralism reviewer.

No demonstrated “close-the-loop” model improvement: the rebuttal strengthens the intended development workflow, but it still does not show a concrete training loop where optimizing the metric yields better pluralism without regressions (e.g., verbosity, false balance).


Automated judge has identifiable systematic bias (e.g., Claude over-rating): they acknowledge it, but it remains a practical limitation for relying on the automated benchmark alone.

**Reviewer Scores:**

Reviewer fPY8: 2 → 4 (borderline / marginal accept)
Likely upward move because the rebuttal directly fixes their main blocker (15 prompts / narrowness) with a 4× expansion (60 prompts, N=1209), but they may still view “no demonstrated improvement loop” as limiting.

Reviewer Zw8e: 4 → 6 (weak accept)
Most of their concrete objections are addressed: novelty/positioning is corrected (no “first” claim), scope is expanded, clustering validity is strengthened with additional analyses, and threshold sensitivity + ablations are added; remaining reservations (US-only, judge bias) keep it from a strong accept.

Reviewer zRFp: 10 → 10 (no change)
Already strong accept; expanded dataset and added analyses would reinforce rather than alter their view.

Reviewer 6u64: 4 → 6 (weak accept)
Breadth concern is largely resolved by the expanded collection; “newer frontier models” is addressed via automated benchmarking; prompt/temperature dependence remains mostly future work, so likely moves to a cautious accept rather than strong.

---

### Decision · Program_Chairs · 2026-01-26

Accept (Poster)